# Micronutrients Deficiency, Supplementation and Novel Coronavirus Infections—A Systematic Review and Meta-Analysis

**DOI:** 10.3390/nu13051589

**Published:** 2021-05-10

**Authors:** Min Xian Wang, Sylvia Xiao Wei Gwee, Junxiong Pang

**Affiliations:** 1Saw Swee Hock School of Public Health, National University of Singapore and National University Health System, Singapore 117549, Singapore; ephwmx@nus.edu.sg (M.X.W.); ephsgxw@nus.edu.sg (S.X.W.G.); 2Centre for Infectious Disease Epidemiology and Research, National University of Singapore, Singapore 117549, Singapore

**Keywords:** micronutrients supplementation, micronutrients deficiency, prevention and treatment, novel coronavirus, COVID-19, SARS, MERS

## Abstract

Background: Micronutrients play roles in strengthening and maintaining immune function, but their supplementation and/or deficiency effects on respiratory tract infections are inconclusive. This review aims to systematically assess the associations between micronutrient supplementation or deficiency, with novel coronavirus incidence and disease severity. Methods: Systematic literature searches conducted in five electronic databases identified 751 unique studies, of which 33 studies (five supplementation studies, one supplementation and deficiency study, and 27 deficiency studies) were eventually included in this review. Proportions of incidence and severity outcomes in each group, and adjusted summary statistics with their relevant 95% confidence intervaIs (CI) were extracted. Data from 19 studies were pooled in meta-analysis using the generic inverse variance method. Findings: A total of 360,346 patients across 16 countries, with a mean age between 32 and 87.7 years, were involved across 33 studies. All studies were on COVID-19 infections. In individuals without micronutrient deficiency, there was a significant reduction on odds of COVID-19 incidence (pooled OR: 0.37, 95% CI: 0.18, 0.78), and ICU admissions or severe/critical disease onset when combined as a severity outcome (pooled OR: 0.26, 95% CI: 0.08, 0.89). Insignificant protective effects were observed on other outcome measures, namely mortality, ICU admission, progression to respiratory-related complications, severe/critical disease onset or requiring respiratory support and hospitalization rate. Conclusion: The absence of micronutrient deficiency significantly reduced COVID-19 incidence and clinical deterioration in hospitalized patients. Usage of micronutrients as prophylaxis and complementary supplement in therapeutic management of COVID-19 patients may be a promising and cost-effective approach warranting in-depth investigation.

## 1. Introduction

Coronaviruses are a family of enveloped RNA viruses capable of infecting both humans and animals [1]. In the past two decades, three beta coronaviruses—severe acute respiratory syndrome coronavirus (SARS-CoV) in 2002, Middle East respiratory syndrome coronavirus (MERS-CoV) in 2012, and SARS-CoV-2 in 2019—with epidemic potential have spilled over from animal reservoirs and affected humans worldwide. Outbreaks caused by SARS-CoV and MERS-CoV were relatively contained, compared to the ongoing COVID-19 pandemic by the SARS-CoV-2. During the eight months where SARS-CoV was actively spreading, the disease infected 8437 people across 29 countries, of which 813 died before the outbreak was declared to be contained globally [2,3]. Since outbreak initiation in April 2012, there were 2566 MERS-CoV cases and 882 associated deaths reported from 27 countries globally. To date, MERS-CoV cases continue to be sporadically reported, mostly in low numbers from the Middle East, while the last known SARS-CoV case was reported in 2004 [4]. In contrast, the COVID-19 pandemic caused by SARS-CoV-2 first discovered in December 2019 has infected at least 119 million people, causing at least 2.6 million deaths across 219 countries as of 12 March 2021 [5]. In comparison to the previous novel coronaviruses, SARS-CoV-2 virus seems to be more transmissible. The relatively higher case rate observed in the COVID-19 pandemic could partially be attributed to a lack of a clear and effective strategy to prevent infection to date. Coupled with no effective strategy to treat COVID-19 cases, the higher morbidity inadvertently led to a relatively higher number of COVID-19 deaths. Remdesivir is the only drug currently approved by the Food and Drug Administration of the United States for treatment [6]. However, it is only recommended as treatment for hospitalised patients requiring supplemental oxygen. There is insufficient data backing any antiviral or antibody therapy for patients with mild to moderate illness.

A robust immune system has general protective effects against disease infection and severity. Micronutrients are shown to be fundamental in strengthening and maintaining immune function [7,8], but the effect of its supplementation and deficiency on respiratory tract infections and pneumonia remains inconclusive [9,10]. Vitamin C and D supplementation or micronutrient sufficiency have been shown to reduce acute respiratory tract infections incidence, including that of influenza, and cold duration, but these observations remain inconsistent [8,9,10,11,12]. Nonetheless, there is sound rationale backing micronutrient supplementation for the prevention of COVID-19 and management of clinical manifestation [13,14]. For instance, vitamin D preserves tight junctions of respiratory cells, kills enveloped virus, and reduces cytokine storm risk, amongst other functions. Zinc is also discovered to inhibit RNA-dependent RNA polymerase activity of SARS-CoV-2. This suggests that the nutritional state of diseased individuals can potentially influence disease prognosis. In the absence of effective prevention and treatment strategies, the potential role of micronutrients protecting against coronavirus infection and prognosis upon infection is a promising and potentially cost-effective approach worthy of investigation.

This review aims to systematically assess the associations between micronutrient supplementation or deficiency, with novel coronavirus incidence and its associated severity. We hypothesize that non-deficiency or supplementation of micronutrients has a protective effect against novel coronavirus incidence and its associated severity.

## 2. Methods

### 2.1. Search Identification and Selection

Relevant peer-reviewed literature that assessed the effect of micronutrient deficiency and/or supplementation on prevention or treatment of a novel coronavirus episode in adults aged above 19 years, were identified and extracted from five electronic databases (Pubmed, EMBASE, Cochrane, Scopus and CINAHL) on 23 October 2020. Specific search terms defined by the Population, Intervention/Exposure, Comparator and Study design (PICOS/PEOS, Appendix A) utilised for each database are provided in Appendix A and the full search strategy used in each database are shown in Appendix A. Reference lists of relevant reviews was also hand-searched to identify additional studies. This study was conducted in accordance with Cochrane’s Preferred Reporting Items for Systematic Reviews and Meta-Analyses (PRISMA) guidelines.

Identified publications were screened according to criteria in the following hierarchy, and included in the review if they fulfilled all criteria:Type of intervention/exposure:Prevention: Supplementation and/or deficiency of any micronutrient singly or in combination with other micronutrients.Treatment: Treatment using any micronutrient singly, or in combination with other micronutrients or therapeutical drugs.Types of outcomes:Prevention: Incidence or episodes of novel coronavirus disease.Treatment: Severity of novel coronavirus disease episode, defined by clinical se-verity, mortality, intensive care unit (ICU) admission, hospitalization duration, or progression to respiratory-related complications.Type of study: Peer-reviewed publications on interventional (randomized controlled trials) and observational (cohort studies, cross-sectional studies and case-controlled studies) studies, with relevant comparator groups, e.g., infected versus non-infected, groups with varying degrees of severity.Type of participants: Adults >19 years, infection-free (prevention) or diagnosed with (treatment) any novel coronavirus disease at time of study recruitment

This review defines novel coronavirus disease as Severe Acute Respiratory Syndrome (SARS), Middle East Respiratory Syndrome (MERS) or Coronavirus Disease-2019 (COVID-19). Extracted studies was first assessed for relevance with titles and abstracts, before retrieval of full texts of relevant studies for further screening and validation based on the above criteria. A PRISMA flow diagram of the study selection process are shown in Figure 1.

### 2.2. Data Extraction

Data extracted from included studies were consolidated with Microsoft Excel 2016, and presented in Table 1, Table 2, Table 3, Table 4 and Table 5. Corresponding authors of included studies were contacted when clarification or more information were required. The following data were extracted from each study: authors, year of publication, study and population characteristics, description of the micronutrient and their relevant categorical cut-off values, and outcomes. Outcome measures extracted when available include (i) number and/or proportion of events, (ii) summary statistic (relative risk (RR), odds ratio (OR), or hazard ratio (HR)) and their corresponding 95% confidence interval (95% CI), and (iii) any other key findings for incidence or severity of the infection in each micronutrient category. As the category definitions and cut-off values were found to vary across micronutrient types, and even within micronutrient types, the relevant data for each category was organised in quintiles, where quintile 1 always refers to the sufficient or non-deficient state for the micronutrient in question. The review authors also noticed that definitions for clinical severity and progression to respiratory-related complications varied across studies, and also extracted the study-specific definitions for these outcomes, when it was reported in the study (Table 5).

The review authors also noticed that some of the selected studies reported the severity outcomes concurrently as a composite outcome and decided to extract this composite outcome (and their study-specific definition) as an outcome for severity on hindsight.

### 2.3. Quality Assessment

Included observational studies were individually evaluated for their methodological quality using methods previously described in another study. The National Heart, Lung and Blood Institute (NHLBI) quality assessment tool for Quality Assessment Tool for Observational Cohort, Cross-sectional and Case-Control Studies [15] tool was utilised to assess the quality of observational studies. The risk of bias in the domains of selection, misclassification, detection, confounding, attrition and inappropriate sample sizes was appraised for each study. The tool did not establish an approach to summarise the overall quality of each assessed study. Thus, the review authors grouped questions into the identified bias domains and assigned the domain-specific quality using the majority of the responses to questions in that domain. A majority of “yes”, “cannot determine”, and “no” responses for questions in the domain corresponded to high, fair and low quality for the domain. In the case of a split response, i.e., equal number of differing responses in the domain, the lower quality is assigned as the domain-specific quality for a conservative judgement. Due to the inability to generate an overall risk of bias assessment, the certainty of evidence for each outcome was not assessed in this review.

Methodological quality of interventional studies were assessed with the Cochrane Risk of Bias Tool 2.0, as according to published instructions [16].

### 2.4. Statistical Analysis

For each outcome, the pooled odds ratio with their corresponding 95% CIs, were estimated with a random-effects model and the generic inverse variance method. In the scenario that the outcome was reported with a mixture of summary statistics and complete data in the 2 × 2 format across studies, crude odds ratios for studies not reporting summary statistics were first estimated using RevMan 5.3. The estimated odds ratios and those reported by the studies were then pooled using the generic inverse variance method for the outcome. Adjusted odds ratios were used to pool summary statistics, when available. As the scale of measurement for clinical severity varied widely across studies (due to study-specific definitions), studies with definitions measuring similar extents of severity (i. guideline classification of clinical condition or need for breathing support and ii. hospitalization rate) for the outcome were pooled together. The progression to respiratory complications outcome also had study-specific definitions. However, the definitions were mainly defined by acute respiratory distress syndrome (ARDS) or requiring oxygen support and/or therapy, and did not vary widely in terms of the scale of severity measurement. Thus, the pooled summary estimate for progression to respiratory-related outcomes were not separated by study-specific definitions.

The I^2^ statistic and Cochran Q test was used to evaluate statistical heterogeneity, where heterogeneity was characterized as minimal (<25%), low (25–50%), moderate (50–75%) or high (>75%) and was significant if *p*-value < 0.05. Subgroup analyses analyzing the effects of: (i) micronutrient type, and (ii) cut-off values for micronutrient categories were also explored. All statistical tests were 2-sided and performed using Review Manager 5.3. Contoured funnel plots assessing publication bias for all pooled outcomes were constructed using the *confunnel* command in STATA 14.2.

All stages of screening, data extraction, and study quality assessments were conducted in duplicate by M.X.W. and S.X.W.G. Discrepancies were resolved by consensus at the end of each procedure before moving on to the next stage of analysis.

## 3. Results

### 3.1. Screening Results and Characteristics of Included Studies

A total of 751 unique studies were screened for relevance, following the removal of 346 duplicates and identification of eight additional studies from external sources. Further assessment of the full-texts of 48 potential studies for eligibility identified a total of 33 studies for inclusion into the systematic review. Due to an inappropriate scale of exposure measurement for pooling (i.e., exposure measured on continuous scale) or outcome measure reported, or lack of sufficient studies with similar outcome definitions, only 19 studies were subsequently included in the meta-analysis (Figure 1). The meta-analysis pooled the following outcomes for studies assessing the effect of micronutrient deficiency: incidence of episodes (five studies), mortality (eight studies), ICU admission (three studies), progression to respiratory-related complications (five studies) and clinical severity according to study-specific definitions (two study-specific definitions, six studies). Studies assessing the effect of micronutrient supplementation were not pooled for any outcome, as outcome measures were reported in an inappropriate format for pooling, or there were insufficient studies with similar outcome definitions reporting the same outcome.

An overview of the study characteristics is presented in Table 1 and Table 2, and details about the population characteristics can be found in Appendix A. All studies were assessing the effect of micronutrient deficiency and/or supplementation on the incidence and/or severity of a COVID-19 episode, despite including SARS and MERS in the search strategy, and were observational in nature. The mean age of 360,346 participants across 16 countries ranged from 32 to 87.7 years old. Most studies were conducted in China (6 studies), United States, United Kingdom, and Austria (three studies each). The largest study was a retrospective study involving 341,484 adults with baseline serum 25-hydroxyvitamin D values from the UK Biobank [17]. Study participants were mainly identified and recruited from hospitals and clinics (28 studies, 12,737 participants), or otherwise from health service registries (three studies, 347,480 participants). There was only 1 study which identified participants amongst institutionalized adults (66 participants) [18], and another study which did not specify the mode of participant identification (63 participants) [19]. Of the 33 studies included in the systematic review, five studies assessed the effect of micronutrient supplementation, 27 studies assessed the effect of micronutrient deficiency, and one study assessed the effect of both micronutrient deficiency and supplementation on the outcomes of interest. Micronutrients assessed for the effect of deficiency included Vitamin D (15 studies), iron (or serum ferritin, 10 studies in total), zinc and calcium (two studies each), and Vitamins B6 and B9 and selenium (one study each). Micronutrients assessed for the effect of supplementation included Vitamin D singly (four studies), or a blend of Vitamins D, B12, and magnesium, and Vitamin C and zinc (one study each).

### 3.2. Methodological Quality of Included Studies

Due to reasons previously mentioned, the overall methodological quality of studies was not generated based on domain-specific assessments. Instead, the qualitative assessments of risk of biases in the domains of selection, misclassification, detection, confounding and others (including inappropriate samples sizes and attrition) are presented in this section. A more detailed breakdown of the domain-specific assessment for each study can be found in Appendix A.

All but three studies were at low risk of selection bias despite inability to determine participation rates of eligible subjected in the studies. The three studies were assessed to be at moderate risk of selection bias as the inclusion criteria and/or definition of cases and controls used to identify and select participants were not clearly prespecified [19,26,43]. Due to inability to assess the blinding of outcome assessors to participants’ exposure status, all studies, except for one study [26], had moderate (29 studies) to high risk of detection bias (two studies). Most studies were with low risk of confounding bias (19 studies), but the remaining which did not measure and adjust for key confounding variables were at high risk of confounding bias (13 studies). All studies were at high risk of bias due to inappropriate samples size, as sample size calculation was not provided in any study, while all but one cohort study [19] were at low risk of attrition bias with high follow-up rates. Cohort and cross-sectional studies all had a low risk of misclassification bias, but case-control studies were at high risk of misclassification bias due to inability to verify that the exposure occurred prior to the outcome.

There were some concerns with the overall risk of bias for the only interventional study included in this review. There were concerns mainly with the risk of bias arising from deviations from the intended interventions and selection of the reported result. These concerns arose mainly from the lack of participant blinding and an unclear analysis plan, which made it difficult to assess if selective reporting was present. The remaining domains relating to selection, detection and attrition bias were at low risk of bias. The domain-specific and overall quality rating given to the study is presented in Appendix A.

The authors also attempted to estimate the presence of publication bias for pooled outcomes, despite the low number of studies included in each outcome (<10 studies, Appendix A). Asymmetry were observed in most plots, and could be due to the low number of studies included in the funnel plots. Nonetheless, the missing studies contributing to asymmetry were mostly in the areas of moderate to high statistical significance in all funnel plots, except that for ICU admission, suggesting that publication bias was not a likely cause for asymmetry in these plots. Missing studies resulting in asymmetry were in the areas of low significance in the funnel plot for ICU admission, but could be due to the low number of studies reporting this outcome (four studies, Appendix AC).

### 3.3. Outcome: Prevention of COVID-19 Episode

A total of 10 studies assessed and reported the effects of Vitamin D deficiency (eight studies) and supplementation (one study), and zinc deficiency (one study) on COVID-19 prevention (Table 3), of which findings from five studies were pooled for meta-analysis (Figure 2). Collectively, non-deficiency in either vitamin D or zinc has a protective effect on COVID-19 incidence, decreasing the incidence odds by 63% (pooled OR: 0.37, 95% CI: 0.18–0.78; Figure 2). High heterogeneity was observed regardless of the micronutrient being assessed (subgroup I^2^ = 84.9%, *p* = 0.01; Figure 2), indicating that the identity of the micronutrient being assessed did not significantly affect incidence odds.

Vitamin D: Differential effects of Vitamin D deficiency (VDD) with COVID-19 incidence were observed. All but two studies reported associations of VDD or insufficiency with increased incidence, or significantly different or lower serum 25(OH)D levels amongst cases compared to non-cases (Table 3). These two studies observed no significant differences in median Vitamin D levels or proportion of cases between participants with and without VDD [32,38]. Nonetheless, Vitamin D supplementation was observed to lower the odds of incidence by 44% amongst free-living patients with Parkinson’s disease (adjusted OR 0.56, 95% CI: 0.32, 0.99) [48]. The pooled OR of 0.49 (95% CI: 0.26, 0.95) from 4 studies also indicated a protective effect of Vitamin D sufficiency on incidence. However, high heterogeneity was observed across the pooled studies (I^2^ = 85%, *p* = 0.0002; Figure 2) even when stratified by their cut-off values for deficiency (I^2^ = 75.1%, *p* = 0.04; Appendix A).

Zinc: Similar to that observed for Vitamin D, significantly lower median zinc levels were reported in cases compared to controls, and the odds of incidence was found to be lowered by 91% when one has sufficient zinc levels (OR: 0.09, 95% CI: 0.03, 0.28) [26].

### 3.4. Outcome: Severity of COVID-19 Episode

Severity of COVID-19 episodes were assessed with a variety of outcome measures across studies. In this review, the generic and study-specific severity outcomes are presented, the manner in which the latter was defined and measured may differ across studies. Generic outcomes include mortality, hospitalization duration and ICU admission, and are presented in Table 4. Outcomes with study-specific definitions include clinical severity, progression to respiratory-related complications and composite outcomes, and are presented in Table 5.

#### 3.4.1. Severity: Mortality from COVID-19

A total of 15 studies observed the effects of micronutrient deficiency (12 studies) and supplementation (3 studies) on deaths from COVID-19 patients. Mortality risk amongst patients supplemented with a bolus dose of Vitamin D within a month before infection was lowered by 89% (HR: 0.11, 95% CI: 0.03, 0.48) [18]. Similarly, supplementation with a vitamin D only (in addition to standard care) or a blend of zinc and vitamin C in patients during their infection resulted in lower proportion of deaths [47], or inverse correlation with mortality amongst inpatients (Pearson correlation coefficient = −0.10, *p*-value not reported) [46]. However, supplementation with a blend of Vitamin D, magnesium and Vitamin B12 during their infection had no observable effects on mortality risk [49].

Micronutrient deficiency had differential effects on mortality risk, but was generally not associated with mortality risk (pooled OR: 0.66, 95% CI: 0.26, 1.66; Figure 3). However, this observation was likely influenced by studies assessing the effect of serum ferritin. Omission of serum ferritin from pooling indicated up to 86% decreased mortality risk with non-deficiency (pooled OR: 0.35, 05% CI: 0.14, 0.89; Appendix A). Nonetheless, high heterogeneity still exists across studies, driven primarily by highly heterogenous findings from studies assessing VDD (overall I^2^ = 84%, *p* < 0.00001; Appendix A). The subgroup heterogeneity even increased after accounting for the different cut-off values defining deficiency and non-deficiency (overall I^2^ = 88%, *p* < 0.0001; I^2^ for 20 ng/mL subgroup = 91%, *p* < 0.00001; Appendix A).

Vitamin D: Non-deficiency in Vitamin D decreases mortality risk, although the reduction was not statistically significant (pooled OR: 0.38, 95% CI: 0.11, 1.34; Figure 3). Studies not included in pooling also indicated similar results, observing higher odds of death with VDD in all patients [37] or lower Vitamin D levels in deceased patients compared to surviving cases and non-cases [31].

Zinc, Selenium: Non-deficiency was not associated to mortality risk (OR: 0.1, 95% CI: 0.01, 1.92; Figure 3) while selenium non-deficiency was associated with decreased mortality risk (OR: 0.35, 95% CI: 0.16, 0.78; Figure 3).

Iron/Ferritin: Non-deficiency had mixed associations with mortality risk, depending on the biomarker used for measuring iron levels. Elevated serum ferritin levels were associated with increased mortality risk (pooled OR 7.48, 95% CI: 2.26, 24.73; Figure 3), but that of post-treatment serum iron levels were associated with lowered mortality odds (adjusted OR: −0.011, 95% CI: −0.019, −0.003; Table 4).

#### 3.4.2. Severity: Hospitalisation Duration from COVID-19

Vitamin D: Deficiency had mixed associations with hospitalization duration, indicating either no association [22] or decreased hospitalization duration in an intermediate care unit [30]. The latter was an interesting observation as patients with severe VDD were found to have poorer outcomes, such as transfer to ICU unit or death, resulting in the relatively shorter stay in the intermediate care unit. Zinc deficiency was associated with longer hospitalization stay, with deficient patients staying approximately 2.2 days longer (*p* = 0.048; Table 4), and 2.39 higher odds of hospitalization exceeding six days (OR: 3.39, 95% CI: 0.99–11.57; Table 4) compared to non-deficient patients [26].

#### 3.4.3. Severity: ICU Admission

Micronutrient deficiency was not associated to ICU admission rates in 5 studies reporting this outcome, although lower ICU admission rates were consistently reported for patients not deficient in Vitamin D or zinc, compared to those who are deficient in these micronutrients (Table 4). It is interesting to note that patients with iron deficiency had relatively lower odds for ICU admission than those without iron deficiency, albeit non-significantly [21]. Nonetheless, the lack of significant association was consistent across the studies regardless of the identity of the micronutrient, when this outcome was assessed qualitatively (Table 4). However, pooled findings from four studies indicate a protective effect of non-deficiency against this outcome (pooled OR: 0.49, 95% CI: 0.29, 0.82; Figure 4). Nonetheless the protective effect with non-deficiency seems to be primarily driven by that of Vitamin D (pooled OR: 0.51, 95% CI: 0.30, 0.87; Figure 4). The protective effect of Vitamin D on ICU admission is further supported by an observed 97% reduction in odds for ICU admission amongst inpatients provided with Vitamin D, in the form of calcifediol, in addition to standard care (adjusted OR: 0.03, 95% CI: 0.003, 0.25) [47]. Changes in the cut-off values defining the Vitamin D sufficiency status do affect the vitamin’s overall effect on ICU admission rates (overall I^2^ = 0%, *p* =0.6; Appendix A).

#### 3.4.4. Severity: Progression to Respiratory-Related Complications

Seven studies assessed the impact of micronutrients on the development of respiratory-related complications. All but two studies [46,49] assessed the impact of micronutrient deficiency on the outcome which encompassed acute respiratory distress syndrome (ARDS) development, need for oxygen therapy and pneumonia due to COVID-19. Non-deficiency in the assessed micronutrients indicated a non-significant protective effect against progression to respiratory-related complications. A further 29% reduction in likelihood of respiratory-related complication progression with non-deficiency, when serum ferritin non-deficiency was excluded from the pooling of effects (pooled OR: 0.88, 95% CI: 0.40, 1.94, Figure 5; pooled OR: 0.59, 95% CI: 0.34, 1.03, Appendix A). The exclusion of serum ferritin also significantly reduces overall heterogeneity from 69% (*p* = 0.002; Figure 5) to 23% (*p* = 0.26; Appendix A). Nonetheless, all suggested protective effects of non-deficiency were not significant. ARDS development was not associated with zinc and VDD (Table 5) [26,30], but significantly higher median serum ferritin levels were observed in patients with ARDS relative to those without ARDS (*p* < 0.001) [41]. The same study found 253% higher odds of ARDS development in patients with heightened serum ferritin levels [41]. Conversely, supplementation with a single micronutrient or a blend of up to three micronutrients, or non-deficient patients were found to correlate inversely with the need for any form of oxygen therapy (Table 5) [46,49].

#### 3.4.5. Severity: Clinical Severity

A total of 14 selected studies reported this outcome, defined by a variety of measures. The most common measure for clinical severity were clinical guidelines published by national or regional health organisations (six studies), followed by hospitalisation, ICU admission or requiring oxygen support (four studies), hospitalisation only (three studies), and onset of symptoms upon infection (two studies). All studies measured clinical severity amongst their subjects with a single measure, except for Ye et al. (2020) which reported clinical severity using clinical guidelines and the number of symptomatic patients [43]. The studies mainly assessed the effect of micronutrient deficiency on clinical severity, including that of Vitamin D (eight studies), iron or ferritin (seven studies), zinc (two studies), and calcium (one study).

Clinical severity was not associated with ferritin, Vitamin D or zinc deficiency regardless of outcome definition used (pooled OR: 0.20, 95% CI: 0.02, 1.55, Figure 6; pooled OR: 0.6, 95% CI: 0.22, 1.65, Appendix A). However, the omission of ferritin indicated a protective effect against all forms of clinical severity defined by clinical guidelines or requiring hospitalisation, ICU admission or respiratory support (pooled OR: 0.08, 95% CI: 0.01, 0.62; Appendix A). When clinical severity was assessed as clinical deterioration, defined by clinical guidelines and ICU admission only, non-deficiency in ferritin, Vitamin D or zinc reduced the odds of severe disease by 74% (pooled OR: 0.26, 95% CI: 0.08, 0.89; Figure 7). The protective effect is even more pronounced when ferritin was omitted from the pooling (pooled OR: 0.16, 95% CI: 0.05, 0.51; Appendix A), although most of the protective effect is driven by non-deficiency in Vitamin D in both forest plots (pooled OR for Vitamin D: 0.18: 95% CI: 0.05, 0.72; Figure 7 and Appendix A). Nonetheless, high heterogeneity exists across the studies pooled, even when observations were stratified according to outcome definitions and micronutrient identity (I^2^ range: 80.5–94%; Figure 6 and Figure 7, Appendix A).

Vitamin D: Three studies reported mixed effects of VDD on COVID-19 associated hospitalisation rates (Table 5), although pooling of two studies indicated no association (OR: 0.60, 95%: 0.22, 1.65; Appendix A). While hospitalisation rate was not continuously associated with 25(OH)D levels (per increase by 10 nmol/L) [17], inpatients had significantly lower mean or median 25(OH)D levels, or a higher VDD prevalence compared to outpatients [33,37]. Similarly, mean 25(OH)D levels eight weeks after disease onset were similar across severity groups, defined by requiring hospitalisation, respiratory support, or intensive care treatment [36].

When clinical severity was defined by clinical guidelines, a non-significant protective effect on severity with Vitamin D sufficiency was observed (pooled OR, 0.10, 95% CI: 0.01, 1.15; Figure 6). In addition, patients with VDD or insufficiency consistently reported to have higher proportions [27,30] or likelihood for severe/critical disease onset (adjusted OR ranging from 1.59 to 14.18) [30,43]. Correspondingly, mild/moderate patients had significantly higher median 25(OH)D levels than severe/critical patients, and a significantly lower proportion of COVID-19 patients with sufficient Vitamin D levels were symptomatic [43].

Iron/Ferritin: Serum ferritin levels were generally higher in patients with more severe conditions, compared to those with a relatively milder disease onset. This observation was consistent across outcome definitions, including clinical guidelines and requiring hospitalisation, ICU admission or requiring oxygen support. However, serum iron levels seem not to be associated with clinical severity, with reportedly similar iron levels between groups with varying disease severity [40,44].

Zinc: Mixed effects were observed in the two studies reporting the effects of zinc deficiency on clinical severity. Symptom onset was seemingly not associated with zinc deficiency [42], although a significantly higher proportion of inpatients with no zinc deficiency require ICU admission, oxygen therapy or respiratory support, compared to those with zinc deficiency, defined by <70 µg/dL zinc [26].

Calcium: Non-deficiency was not associated to clinical severity as mean total or ionized calcium levels eight weeks after disease onset were similar across the mild, moderate and severe groups, defined by requiring hospitalisation, respiratory support, or intensive care treatment [36].

#### 3.4.6. Severity: Composite Outcomes

The effect of micronutrient deficiency and supplementation on composite outcomes was assessed in four studies (Table 5). Composite outcome entailed death, ICU admission and/or oxygen therapy. Patients insufficient or deficient in calcium and Vitamin D consistently indicated higher odds/hazards of composite outcome (ranging from 1.96 to 5.12 times) compared to those not insufficient or deficient in five study arms across three studies [28,29,37]. In contrast, mixed observations were observed when patients were supplemented with Vitamin D singly or a blend of Vitamin D, Magnesium, and Vitamin B12 in two study arms in two studies [29,49]. Between patients with and without the composite outcome, Vitamin D supplementation status did not differ significantly (*p* > 0.05) [29]. However, composite outcome incidence was significantly higher amongst patients without micronutrient blend supplementation (*p* = 0.006) [49]. In terms of median micronutrient level between patients with and without composite outcome, a significantly lower median calcium level was observed in patients in composite outcome (2.01 mmol/L (1.97–2.05) vs 2.10 mmol/L (2.03–2.20), *p* < 0.001) [28], while Vitamin D levels were similar across patients with and without the composite (*p* > 0.05) [29] (Table 5).

## 4. Discussion

This review sought to provide a comprehensive overview of literature investigating association between any micronutrient deficiency or supplementation and the three beta coronaviruses. However, only literature on COVID-19 induced by SARS-CoV-2 were recovered from the systematic search. Our findings demonstrate significantly reduced odds of COVID-19 incidence (pooled OR: 0.37, 95% CI: 0.18, 0.78), and ICU admissions or severe/critical disease onset (pooled OR: 0.26, 95% CI: 0.08, 0.89) in individuals without micronutrient deficiency. Similar protective effects, albeit insignificant, were also observed for amongst individuals without any micronutrient deficiency for the following outcomes: mortality (pooled OR: 0.66, 95% CI: 0.26, 1.66), ICU admission (pooled OR: 0.57, 95% CI: 0.3, 1.07), progression to respiratory-related complications (pooled OR: 0.59, 95% CI: 0.34, 1.03), severe/critical disease onset or requiring respiratory support (pooled OR: 0.20, 95% CI: 0.02, 1.55), and hospitalisation rate (pooled OR: 0.60, 95% CI: 0.22, 1.65).

Vitamin D was the most studied micronutrient in this review, and its beneficial effect of Vitamin D on the immune system maintenance and enhancement has been demonstrated [50]. However, evidence on its role in other health and disease remains inadequate and recent guidelines published by the National Institute for Health and Care Excellence concluded insufficient evidence to recommend Vitamin D supplementation to prevent or treat COVID-19 [51,52]. Apart from supplementation trial results [53], there are two points worthy of consideration in the debate on vitamin D in COVID-19. Seasonal variations of vitamin D levels, which trough during winter, have mirrored seasonal variations of respiratory viral infections, suggesting an association between the two [54,55]. Furthermore, groups with higher prevalence of VDD, such as elderly, Black, Asian, and minority ethnic populations, are disproportionately affected by COVID-19 [56,57,58,59]. This could be attributed to socio-economic differences and the age gradient in clinical progression from COVID-19 disfavoring older people, but also suggests an association between vitamin D deficiency, the immune functioning and infection rates. Analysis of COVID-19 deaths and data in 20 European countries after their peak had similar findings with this review. The study found inverse correlations between mean Vitamin D levels and COVID-19 incidence rates (Pearson’s correlation r = −0.477, *p* = 0.033), although that for mortality rates did not achieve statistical significance (r = −0.357, *p* = 0.123) [60]. Low Vitamin C levels were also observed in higher prevalence amongst critically ill hospitalized patients with respiratory infections, pneumonia, sepsis and COVID-19 [61].

The scientific rationale backing the supposed efficacy of micronutrient supplementation to prevent or treat COVID-19 have been documented [13,14], but primary evidence on it remains sparse. Vitamin C administered orally (2–8 g/day) may reduce incidence and length of respiratory infections. When administered intravenously (6–24 g/day), Vitamin C reduces mortality and severe disease progression in terms of duration spent in hospital, ICU, or on mechanical ventilation [61]. Hemila’s meta-analysis of controlled clinical trials further demonstrated a reduced length of stay in the intensive care unit and duration of mechanical ventilation with oral or intravenous vitamin C [12]. Nonetheless this review only identified a single study on vitamin C, which examined the effect of vitamin C in combination with zinc. Thus, more detailed investigation is required to understand the potential and isolated effect of Vitamin C. Likewise, only three studies investigated zinc, selenium, calcium and vitamin B [25,26,42]. Of these micronutrients, zinc has garnered much interest due its dual-immunomodulatory and anti-viral properties. At the time of writing, there are 42 registered clinical trials investigating the combinatorial use of zinc with anti-viral drugs or other micronutrients as supplementation [62]. Included studies investigated zinc cut-offs of 70–80 μg/dL, aligning with cut-off values defined from a nationally representative survey [63].

Sensitivity analysis on the outcome was conducted to exclude studies examining ferritin. In all cases, the protective effect of non-deficiency increased following exclusion as seen from a reduction in odds of adverse outcomes–mortality (pooled OR: 0.35, 95% CI: 0.14, 0.89; Appendix A), progression to respiratory-related complication (pooled OR: 0.59, 95% CI: 0.34, 1.03; Appendix A) and clinical severity (pooled OR: 0.08, 95% CI: 0.01, 0.62; Appendix A). The protective effect of micronutrient non-deficiency on mortality and study-defined clinical severity became significant with the omission of ferritin during pooling. Ferritin is an iron storage protein that is commonly used to evaluate iron stores and diagnose iron deficiency. However, the use of ferritin as a standalone gauge of iron load is precarious as it is also an inflammatory marker [64]. Thus, it is unclear whether serum ferritin reflects or holds a role in disease severity. As serum ferritin correlate with both iron store and disease, the World Health Organization recommends adjusting for inflammation by applying correction factors (acute phase response proteins CRP and AGP) concurrently assessed with ferritin, or raising the cut-off defining deficiency [65].

The absence of literature exploring the relationship between micronutrients and the earlier novel coronaviruses is likely due to different outbreak trajectories caused by fundamental differences in viral tropism, dynamics and transmission, despite their higher case fatality rates. SARS was brought under control swiftly with intense public health measures in 2003 while MERS continues to record predominately zoonotic cases to this date. Both coronaviruses had tropism for the lower respiratory tract that manifested in severe clinical progress more frequently, resulting in relatively higher case fatality rates (case-fatality rate: SARS 9.7%, MERS 34%, COVID-19 2.2%) [66,67]. However, this also translated to less community transmission and higher rates of nosocomial transmission over time. This contrasts SARS-CoV-2′s tropism for upper respiratory tract cells and viral load that peaks and decreases within a much shorter timeframe, facilitating ease of community spread when symptoms are mild or absent [66,67]. With a better understanding of the impact of micronutrients deficiency on susceptibility to novel coronavirus infection and severity, encourages policy makers to focus on populations potentially at risk of micronutrient deficiency, such as low-income groups, migrants, refugees and minority ethics groups, to minimize active transmission of novel coronaviruses, especially when vaccines may not be prioritized for them.

### Strengths and Limitations

This is the first systematic review assessing the effect of micronutrients collectively on all novel coronavirus incidence and severity, to the best knowledge of the authors. Comparatively, existing reviews only assessed the effect of a specific micronutrient, such as Vitamin C or Vitamin D on a single coronavirus, i.e., SARS-CoV-2 [60,61,68]. In addition, there was no restriction on publication year during the search, thus all relevant literature published since the inception of the databases should have been included. Collectively, these increase the relative comprehensiveness of the overview presented in this review. Studies included in this review spanned across 16 countries, of which 10 were in Asian populations. This increases the external validity to the outcomes reported in this review, to Asian and non-Asian populations.

Limitations also exist in this review. As mentioned, the studies included in this review are only limited to COVID-19. This decreases the generalizability of micronutrient’s preventative and therapeutic effect to other novel coronaviruses, regardless those in the past or future. In addition, this review only included 1 randomized controlled trial investigating micronutrient as a therapeutic agent, the gold standard to establish causal relationship in scientific evidence. Remaining studies included were observational in nature —most observed for presence of deficiency in patients, and only four studies were supplementation trials that may also not involve any form of intervention. Thus, more evidence from clinical trials is required to provide robust guidance on the usage of micronutrients and its respective feasibility, dosage and regimen to prevent or treat coronavirus episodes. Moreover, the results from this review may only represent the combined effect of all micronutrients for each outcome in its entirety. This review was unable to discuss the stratified effects any micronutrient singly, should there only be a single study for that micronutrient due to concerns about statistical accuracy. In addition, the spectrum of micronutrients investigated skewed towards vitamin D. At least half of the studies included (19 studies, 57.6%) investigated vitamin D singly or as one of multiple micronutrients in question. This was followed by ferritin/iron with eight studies. Given the plethora of micronutrients with beneficial roles in the immune system, more research is warranted on the other micronutrients to examine their synergy. Furthermore, we observed inconsistent definitions for “deficiency” and “insufficiency”, with cut-off values ranging from 0.2 ng/mL to 30 ng/mL. The Endocrine Society Taskforce defined a cut-off at 20 ng/mL (50 nmol/L) for severe vitamin D deficiency, aligned with most expert bodies and societies, including the Institute of Medicine (IOM) [69]. The adoption of a uniform cut-off allows for consistency in evidence investigating a relationship between vitamin D levels and disease. Nonetheless, cut-off values should be adjusted considering differences in BMI and vitamin D metabolism across populations, given the varied prevalence of vitamin D deficiency across ethnic groups and geographical regions. Lastly, the results from this review should be interpreted with caution due to the generally low methodological quality of included studies, and the inability to assess the certainty of evidence for each outcome due to the inability to assess the overall risk of bias. Included studies were mostly at high risk of detection bias and may not have sufficient statistical power, as all but one study lacked sample size calculations.

## 5. Conclusions

This review showed that micronutrients in their entirety have effects on COVID-19 incidence and severity outcomes. Individuals without micronutrient deficiency had reduced odds of COVID-19 incidence and disease severity. Integrating micronutrients into the prevention and therapeutic management of COVID-19# may complement non-pharmaceutical interventions to reduce the risk of transmission and disease severity in an unvaccinated population.

## Figures and Tables

**Figure 1 nutrients-13-01589-f001:**
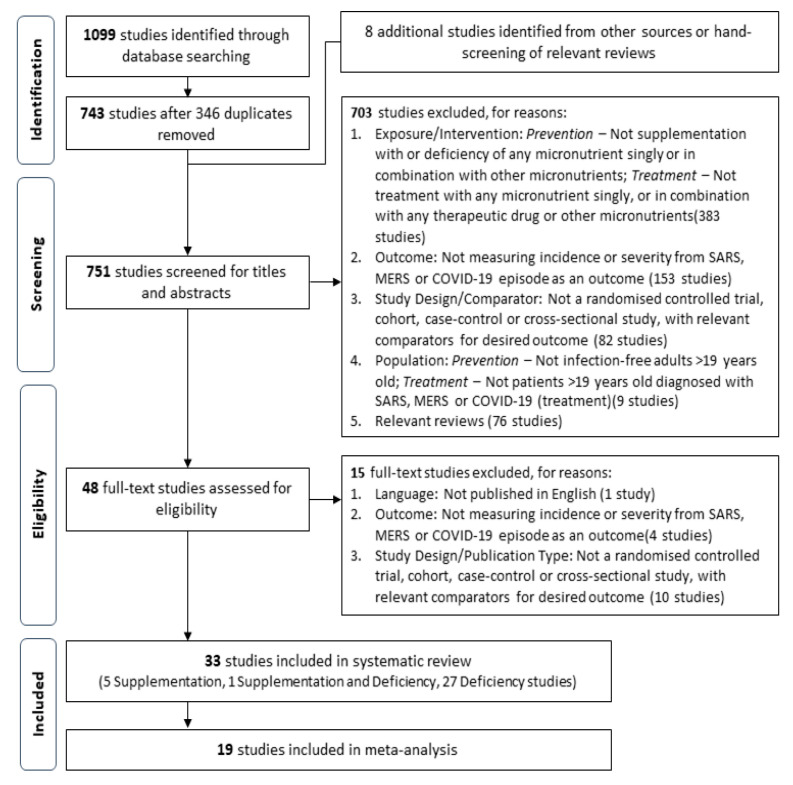
Flowchart of the screening and study selection process.

**Figure 2 nutrients-13-01589-f002:**
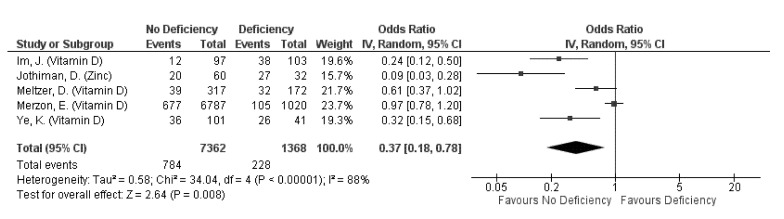
Pooled odds ratio on the incidence of COVID-19 episode when subjects are not deficient, compared to those deficient in any micronutrient singly.

**Figure 3 nutrients-13-01589-f003:**
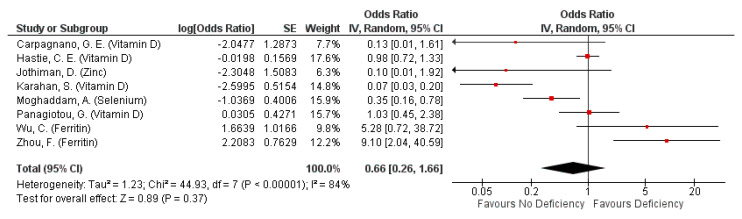
Pooled odds ratio on mortality from COVID-19 episode, or during hospitalisation due to COVID-19 episode when hospitalized subjects are not deficient, compared to those deficient in any micronutrient singly.

**Figure 4 nutrients-13-01589-f004:**
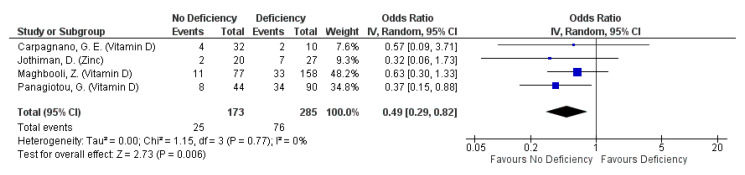
Pooled odds ratio on ICU admission due to COVID-19 episode when hospitalized subjects are not deficient, compared to those deficient in any micronutrient singly.

**Figure 5 nutrients-13-01589-f005:**
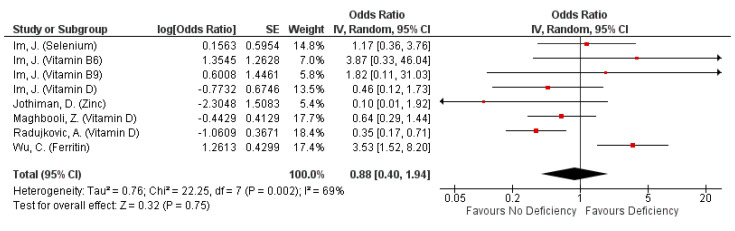
Pooled odds ratio on progression to respiratory-related complications due to COVID-19 episode when hospitalized subjects are not deficient, compared to those deficient in any micronutrient singly.

**Figure 6 nutrients-13-01589-f006:**
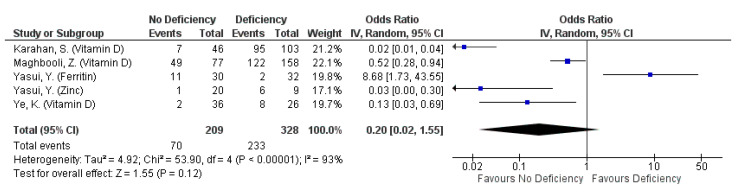
Pooled odds ratio on clinical severity of COVID-19 patients (defined by clinical condition, or need for breathing support) hospitalized subjects are not deficient, compared to those deficient in any micronutrient singly.

**Figure 7 nutrients-13-01589-f007:**
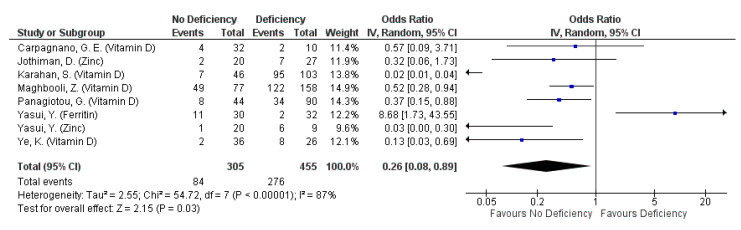
Pooled odds ratio on clinical deterioration of COVID-19 patients (defined by severe/critical clinical condition or ICU admission) hospitalized subjects are not deficient, compared to those deficient in any micronutrient singly.

**Table 1 nutrients-13-01589-t001:** Studies assessing the micronutrient deficiency on outcomes.

Author, Year [Ref.]	Reported Study Design, Country of Study	Population Source: Period of Data Collection	Population Size (% Men); # Subjects with ≥1 Comorbidity	Mean Age in Years (SD)	Micronutrient in Question	Blood Sampling Timepoint	Micronutrient Categories, as Defined by Study: Cut-off for Each Category, as Defined by Study			Outcome Reported [I: Incidence of COVID-19 Episode; S, Severity of COVID-19 Episode]
							Quintile 1	Quintile 2	Quintile 3	
Baktash, 2020 [20]	Prospective cohort; UK	Hospital: 1 Mar–30 Apr 2020	105 (54.3); 54	Cases (deficient): 79.46 (89.52) Cases (non-deficient): 81.16 (7.23) Controls: 83.44 (8.08)	Vitamin D	At admission	NA: measured as a continuous outcome			I: Incidence of COVID-19 episode
Bellmann-Weiler, 2020 [21]	Retrospective cohort; Austria	Hospital: 25 Feb–20 May 2020	259 (60.6);152	68 (53–80) *	Iron	At admission (Day 1 ± 1)	No Iron deficiency (ID)	Absolute ID: TSAT <20% + Serum ferritin <100 µg/L	Functional ID: TSAT <20% + Serum ferritin >100 µg/L	S: Death during hospitalisation from COVID-19 episode S: ICU admission due to COVID-19 episode
Carpagnano, 2020 [22]	Retrospective cohort; Italy	Hospital: 11 Mar–30 Apr 2020	42 (71.4); 36	≥30 ng/mL: 64 (18) 30 >Vitamin D ≥20 ng/mL: 64 (13) 20 > Vitamin D ≥10 ng/mL: 60 (6.9) <10 ng/mL: 74 (11)	Vitamin D	≤12 h following RICU admission	NR: ≥10 ng/mL	NR: <10 ng/mL	-	S: Death during hospitalisation from COVID-19 episode S: Hospitalisation duration due to COVID-19 episode S: ICU admission due to COVID-19 episode
D’Avolio, 2020 [23]	Retrospective cohort; Switzerland	Hospital: 1 Mar–14 Apr 2019, 1 Mar–14 Apr 2020	107 (54.2); NR	73 (63–81) *	Vitamin D	<7 weeks after PCR result, overall median days from result = 3.0 (IQR:0.0 to 7.0)	NA: measured as a continuous outcome			I: Incidence of COVID-19 episode
		NR (controls): 1 March to 14 April 2019	1377 (45.3); NR	63 (46–76) *						
Dahan, 2020 [24]	Cross-sectional; Israel	Hospital: 21 Feb–30 Mar 2020	39 (59); 22	52.46 (2.76)	Iron (Serum ferritin)	On admission	NA: measured as a continuous outcome			S: Clinical severity of COVID-19 episode at admission (defined by Report of the WHO-China Joint Mission)
Hastie, 2020 [17]	Retrospective cohort; UK	Health registries: 5 Mar–25 Apr 2020	341,484 (NR); NR	37–73 (between 2006–2010) ^#^	Vitamin D	Baseline measurement taken between 2006–2010	Sufficient: ≥50 nmol/L	Insufficient: <50 nmol/L	Deficient: <25 nmol/L	S: Death from COVID-19 episode S: Hospitalisation rate due to COVID-19 infection
							NA: measured as a continuous outcome (per 10 nmol/L)			
Im, 2020 [25]	Case-control; South Korea	Hospital: Feb–Jun 2020	Cases: 50 (42); NR Controls: 150 (sex-matched to cases); NR NR	Cases: 57.5 (34.5–68) Controls: NR. but age-matched to controls	Vitamin D3	≤7 days of admission (median number of days from admission = 2 days)	No deficiency	Deficiency: ≤20 ng/dL	Severe Deficiency: ≤10 ng/dL	I: Incidence of COVID-19 episode S: Progression to respiratory related complication due to COVID-19 episode (defined by pneumonia incidence, or requiring high-flow nasal cannula, mechanical ventilator, and extracorporeal membrane oxygenation or death).
					Vitamin B6			Deficiency: ≤5 µg/L	-	S: Progression to respiratory related complication due to COVID-19 episode (defined by pneumonia incidence, or requiring high-flow nasal cannula, mechanical ventilator, and extracorporeal membrane oxygenation or death).
					Vitamin B9			Deficiency: ≤4 µg/L	-	
					Selenium			Deficiency: ≤95 µg/L	-	
					≥1 deficiency			Any of the above	-	
Jothimani, 2020 [26]	Prospective case-control; India	Hospital: 17 May–27 May 2020	Cases: 47 (61.5); NR Controls: 45 (67.7); 0	Cases: 34 (18–77) * Controls: 32 (18–60) *	Zinc ^a^	6 h from admission	No deficiency	Deficiency: ≤80 µg/L	-	I: Incidence of COVID-19 episode S: Death during hospitalisation from COVID-19 episode S: Hospitalisation duration due to COVID-19 infection S: ICU admission due to COVID-19 episode S: Progression to respiratory related complication due to COVID-19 episode (defined by ARDS development)
Karahan, 2020 [27]	Retrospective cohort; Turkey	Hospital: 1 Apr–20 May 2020	149 (54.4); 85	63.5 (15.3)	Vitamin D3	NR	Normal: ≥ 30 ng/mL	Insufficiency: 21–29 ng/mL	Deficiency: ≤20 ng/mL	S: Death during hospitalisation from COVID-19 episode S: Clinical severity of COVID-19 episode (defined by Chinese Clinical Guideline)
Liu, 2020 [28]	Retrospective cohort; China	Hospital: 9 Feb–15 Feb 2020 (follow-up till 25 Feb 2020)	107 (49); 40	68 (61–76) *	Calcium (Serum Calcium)	≤24 h from admission	Normal: 2.15–2.5 mmol/L	Hypocalcemia: <2.15 mmol/L	-	S: Composite outcome for need for mechanical ventilation, ICU admission due to COVID-19 episode, or all-cause mortality during admission
Macaya, 2020 [29]	Retrospective cohort; Spain	Hospital: 5 Mar–31 Mar 2020	80 (43.8); 50	non-severe patients: 63 (50–72) * severe patients: 75 (66–84) *	Vitamin D3	At admission or ≤3 months before admission	No deficiency: ≥20 ng/mL	Deficiency: <20 ng/mL	-	S: Composite outcome for death, ICU admission or requiring high flow oxygen (greater than nasal cannula) due to COVID-19 episode
Maghbooli, 2020 [30]	Cross-sectional; Iran	Hospital: Till 1 May 2020 (start date unspecified)	235 (61.3); NR	58.72 (15.22)	Vitamin D3	At admission	No deficiency: ≥30 ng/mL	Deficiency/ Insufficiency: <30 ng/mL	-	S: Clinical severity of COVID-19 episode (defined by Chinese Clinical Guideline) S: Hospitalisation duration due to COVID-19 infection S: ICU admission due to COVID-19 episode S: Progression to respiratory related complication due to COVID-19 episode (defined by ARDS development)
Mardani, 2020 [31]	Cross-sectional; Iran	Medical Center: Mar 2020	123 (52.8); NR	42 (NR)	Vitamin D	At admission	NA: measured as a continuous outcome			I: Incidence of COVID-19 episode S: Death from COVID-19 Episode
Meltzer, 2020 [32]	Retrospective cohort; United States	Hospital: 3 Mar–10 Apr 2020	489 (25); 261	49.2 (18.4)	Vitamin D	Baseline measurement taken within 1 year to 14 days before patient’s COVID-19 test	No deficiency: ≥20 ng/mL OR ≥18 pg/mL	Deficiency: <20 ng/mL	-	I: Incidence of COVID-19 episode
Merzon, 2020 [33]	Retrospective cohort; Israel	Medical Center: 1 Feb–30 Apr 2020	7807 (41.4); 2136	COVID-19 positive: 35.58 (0.56) COVID-19 negative: 47.35 (0.24)	Vitamin D	NR	Sufficiency: ≥30 ng/mL	Insufficiency: 29–20 ng/mL	Deficiency: <20 ng/mL	I: Incidence of COVID-19 episode
							Normal: ≥30 ng/mL	Low: < 30 ng/mL	-	
							NA: measured as a continuous outcome			I: Incidence of COVID-19 episode S: Hospitalisation rate due to COVID-19 episode
Moghaddam, 2020 [34]	Cross-sectional; Germany	Hospital: NR	33 (42.4); 22	77 (38–94) *	Selenium (Serum Selenium)	Throughout hospitalisation; mean (SD) samples drawn per patient = 5.03 (4.27)	Normal: 45.7–131.6 μg/L	Deficiency: <45.7 μg/L	-	S: Death during hospitalisation from COVID-19 episode
							NA: measured as a continuous outcome			
Panagiotou, 2020 [35]	Cross-sectional; UK	Hospital: NR	134 (54.5); 114	severe: 61.1 (11.8) mild: 76.4 (14.9)	Vitamin D	NR	Normal: ≥50 nmol/L	Deficient: <50 nmol/L	-	S: Death during hospitalisation from COVID-19 episode S: Clinical severity of COVID-19 episode (defined by admission to intensive therapy unit)
Pizzini, 2020 [36]	Prospective cohort; Austria	Medical Center: From 29 Apr 2020 (end date unspecified)	109 (60); 88	58 (14)	Vitamin D	8 weeks after COVID-19 diagnosis	NA: measured as a continuous outcome			S: Clinical severity of COVID-19 episode (defined by study criteria requiring hospitalisation, respiratory support or intensive care treatment)
					Calcium (Total, ionised)		NA: measured as a continuous outcome			
					Iron (Serum ferritin)		NA: measured as a continuous outcome			
Radujkovic, 2020 [37]	Prospective cohort; Germany	Hospital: 18 Mar–18 Jun 2020	185 (51); 77	60 (49–70) *	Vitamin D	At admission and SARS-CoV-2 testing	No deficiency: ≥12 ng/mL	Deficiency: <12 ng/mL (<30 nM)	-	S: Death during hospitalisation from COVID-19 episode (all-cause mortality) S: Progression to respiratory-related complications (defined by requiring any form of oxygen therapy) S: Hospitalisation rate from COVID-19 episode S: Composite event of mechanical invasive ventilation and/or death from COVID-19 episode
							No insufficiency: ≥20 ng/mL	Insufficiency: <20 ng/mL	-	S: Death during hospitalisation from COVID-19 episode (all-cause mortality) S: Composite event of mechanical invasive ventilation and/or death from COVID-19 episode
							NA: measured as a continuous outcome			S: Hospitalisation rate from COVID-19 episode
Raisi-Estabragh, 2020 [38]	Prospective cohort; UK	Health registries: 16 Mar–18 May 2020	4510 (48.8); 2081	COVID-19 positive: 68.11 (9.23) COVID-19 negative: 68.91 (8.72)	Vitamin D	Baseline measurement taken between 2006–2010	NA: measured as a continuous outcome			I: Incidence of COVID-19 episode
Smith, 2020 [39]	Retrospective multi-centre cohort; United States	Hospital: 1 May–30 Mar 2020	86 (0); 86	68.5 (59–74.8)	Iron (Ferritin)	At admission	NA: measured as a continuous outcome			S: Clinical severity of COVID-19 episode (defined by study criteria of hospitalisation and/or ICU admission, requiring mechanical ventilation and/or death)
Sonnweber, 2020 [40]	Prospective multi-centre cohort; Austria	Hospital: NR	109 (60); 88	58 (14)	Iron (Ferritin)	60 days (SD ± 12) after the onset of first COVID-19 symptoms	NA: measured as a continuous outcome			S: Clinical severity of COVID-19 episode (defined by study criteria of ICU admission, requiring oxygen therapy or respiratory support)
Sun, 2020 [19]	Retrospective cohort; China	NR: NR	63 (58.7); 12	Median: 47 (Range: 3–85)	Iron (Serum Ferritin)	On admission	NA: measured as a continuous outcome			S: Clinical severity of COVID-19 episode (defined by New Coronavirus Pneumonia Prevention and Control Program, 7th edition)
Wu, 2020 [41]	Retrospective cohort; China	Hospital: 25 Dec 2019–26 Jan 2020 (follow-up till 13 Feb 2020)	201 (63.7); 39	51 (43–60) *	Iron (Serum Ferritin)	≤24 h from admission	NR: >300 ng/mL	NR: ≤300 ng/mL	-	S: Death from COVID-19 episode S: Progression to respiratory-related complication due to COVID-19 episode (defined by ARDS development)
							NA: measured as a continuous outcome			
Yasui, 2020 [42]	Retrospective cohort; Japan	Health Center: 24 Mar–24 May 2020	62 (54.8);	NR, but 17 (27.4%) are aged ≥65 years	Iron (Ferritin)	Multiple timepoints: on first day of hospitalisation and 2–3 days later on an empty stomach in the early morning after hospitalisation	NR: ≥300 ng/mL (male), ≥200 ng/mL (female)	NR: <300 ng/mL (male), <200 ng/mL (female)	-	S: Clinical severity of COVID-19 episode (defined by study criteria of ICU admission, requiring oxygen therapy or respiratory support)
							NA: measured as a continuous outcome			
					Zinc		NR: ≥70 µg/dL	NR: <70 µg/dL	-	
							NA: measured as a continuous outcome			
Ye, 2020 [43]	Case-control; China	Hospital: 16 Feb–16 Mar 2020	Cases: 62 (37); 16 Controls: 80 (42); 0	Cases: 43 (32–59) * Controls: 42 (31–52) *	Vitamin D	At admission	Sufficiency: ≥75 nmol/L	Insufficiency: 50 nmol/L ≤25(OH)D <75 nmol/L	Deficiency: <50 nmol/L	S: Clinical severity of COVID-19 episode (defined by Chinese National Health Commission Guidelines (6th edition))
							Non-deficiency: ≥50 nmol/L	Deficiency: <50 nmol/L	-	I: Incidence of COVID-19 episode S: Clinical severity of COVID-19 episode (defined by Chinese National Health Commission Guidelines (6th edition))
							NA: measured as a continuous outcome			
Zhao, 2020 [44]	Retrospective cohort; China	Hospital: 1 Feb–29 Feb 2020	50 (60); 17	55 (44–66) *	Iron (Pre- and post-treatment serum iron)	NR	NA: measured as a continuous outcome			S: Death from COVID-19 episode S: Clinical severity of COVID-19 episode (defined by Chinese National Health Commission Guidelines (7th edition))
Zhou, 2020 [45]	Retrospective cohort; China	Hospital: 29 Dec 2019–31 Jan 2020	191 (62.3); 91	Survivor: 52 (45–58) * Non-survivor: 69 (63–76) *	Iron (Serum Ferritin)	NR: frequency of examinations determined by treating physician	NR: >300 ng/mL	NR: ≤300 ng/mL	-	S: Death from COVID-19 episode
							NA: measured as a continuous outcome			

NA, Not Applicable; NR, Not Reported; UK, United Kingdom; CKD, chronic kidney disease; CVD, cardiovascular disease; COPD, chronic obstructive pulmonary disease; ARDS, acute respiratory distress syndrome. ***** median (IQR); ^#^ range. ^a^ All patients received hydroxychloroquine, antibiotics, and multivitamins, including vitamin C 500 mg twice a day and zinc 150 mg once a day (after the test).

**Table 2 nutrients-13-01589-t002:** Studies assessing micronutrient supplementation on outcomes.

Author, Year [Ref.]	Reported Study Design; Country of Study	Population Source: Period of Data Collection	Population Size (% Men); Number of Subjects with ≥1 Comorbidity	Mean Age (SD)	Micronutrient in Question	Outcome Reported [I: Incidence of COVID-19 Episode; S, Severity of COVID-19 Episode]
Annweiler, 2020 [18]	Retrospective cohort; France	Nursing Home: Mar–Apr 2020	66 (22.7); 66	87.7 (9.0)	Vitamin D3 ^a^	S: Death from COVID-19 episode
Capone, 2020 [46]	Retrospective cohort; United States	Hospital: till 20 Apr 2020 (start date unspecified)	102 (53.9); 61	63.22 (53.3–74.3) *	Vitamin C & zinc ^b^	S: Death during course of follow-up (all-cause mortality) S: Progression to respiratory-related complication (defined by requiring invasive mechanical ventilation) due to COVID-19 episode
Castillo, 2020 [47]	Open-label, double blind randomised controlled trial; Spain	Hospital: Unspecified	76 (58.2); 26	53 (10)	Vitamin D3 ^c^	S: Death during hospitalisation from COVID-19 episode S: ICU admission
Fasano, 2020 [48]	Retrospective, single-center case-control; Italy	Health registry: NR	1486 (56.9); 1486	Case: 70.5 (10.1) Controls: 73.0 (9.5)	Vitamin D ^b^	I: Incidence of COVID-19 episode
Macaya, 2020 [29]	Retrospective cohort; Spain	Hospital: 5 Mar–31 Mar 2020	80 (43.8); 50	non-severe patients: 63 (50–72) * severe patients: 75 (66–84) *	Vitamin D ^b^	S: Composite outcome for death, ICU admission and/or need for higher oxygen flow than that provided by a nasal cannula due to COVID-19 episode
Tan, 2020 [49]	Retrospective cohort; Singapore	Hospital: 15 Jan–15 Apr 2020	43 (60.4); 24	Non-supplemented: 64.1 (7.9) Supplemented: 58.4 (7)	Vitamin D, Magnesium & Vitamin B12 ^d^	S: Death during hospitalisation from COVID-19 episode S: Composite outcome for progression to respiratory-related complication (defined by requiring oxygen therapy) or ICU admission due to COVID-19 episode S: Progression to respiratory-related complication (defined by requiring oxygen therapy)

NR, Not Reported. ***** median (interquartile range). ^a^ Oral supplementation of a 80,000 IU bolus dose in the week after/just before diagnosis, or in the previous month. ^b^ Supplementation status treated as a ordinal variable (Yes/No). ^c^ Oral Calcifediol in soft capsules (0.532 mg); Patients in the calcifediol treatment group continued with oral calcifediol (0.266 mg) on day 3 and 7, and then weekly until discharge or ICU admission. ^d^ Oral supplementation of 1000-IU dose of vitamin D3 (cholecalciferol), 150 mg of magnesium oxide, and 500 mg vitamin B12 (methylcobalamine) for ≤14 days daily, until patient subsequently deteriorated or was deemed to have recovered based on symptom resolution and two consecutive negative SARS CoV-2 reverse transcriptase PCR respiratory sample.

**Table 3 nutrients-13-01589-t003:** COVID-19 Incidence as outcome.

Author [Ref]	Micronutrient in question	Odds Ratio (OR) (95% Confidence Interval); % Population Infected (Infected/Population Size)					Key Findings
		Supplementation		Deficiency			
		Supplemented	Non-supplemented	Quintile 1	Quintile 2	Quintile 3	
Baktash [20]	Vitamin D						**Significantly lower vitamin D levels** in the cases ˠ (27.00 nmol/L) compared to non-cases ˠ (52.00 nmol/L, *p* = 0.0008).
D’Avolio [23]	Vitamin D						**Significantly lower 25(OH)D levels** in cases (11.1 ng/mL) to non-cases (24.6 ng/mL, *p* = 0.004). **Significant difference in 25(OH)D levels** in cases ˠ compared with the 2019 non-cases ˠ (24.6 ng/mL, *p* < 0.001); but no difference between the 2019 and 2020 non-cases ˠ (*p* = 0.076).
Fasano [48]	Vitamin D	0.56 (0.32- 0.99) *^,a^; 4.0% (13/329)	Reference; 8.0% (92/1157) *				**44% ^a^ lowered odds of incidence** when patients reported vitamin D supplementation, compared to non-supplementation (*p* = 0.048).
Im [25]	Vitamin D3			NR; 12.4% (12/97) ^#^	NR; 32.5% (26/80) ^#^	NR; 52.2% (12/23) ^#^	**Significant difference in proportions of cases and controls** with deficiency (*p* = 0.003) and severe deficiency (*p* = 0.001). **Significantly lower mean 25(OH)D levels** in cases (15.7 ± 7.9 ng/dL) than in controls (25.0 ± 13.2 ng/dL, *p* < 0.001).
Mardani [31]	Vitamin D						**Significantly lower mean Vitamin D levels** in to cases ˠ (18.5 ng/mL) than in non-cases ˠ (30.2 ng/mL, *p* < 0.0001).
Meltzer [32]	Vitamin D			NR; 12.3% (39/317)	NR; 18.6% (32/172)		**No difference** in proportion of cases ˠ and non-cases ˠ between non-deficient and deficient groups (*p* = 0.06).
Merzon [33]	Vitamin D ᶧ			Reference; 69.4% (79/1139) ^#^	1.59 (1.29–2.02); 10.6% (598/5648) ^#^	1.58 (1.13–2.09); 10.3% (105/1020)^#^	**58% to 59% ^b^ increase in odds of incidence** for the deficient and insufficient group respectively, compared to those with sufficient vitamin D levels. **50% ^b^ increase in odds of incidence** for the those with low vitamin D levels, compared to those with normal vitamin D levels. **Significantly lower mean (****±****SD) vitamin D levels** in cases ˠ (19 ± 8.42 ng/mL) than non-cases ˠ (20.55 ± 9.84 ng/mL, *p* = 0.026).
	Vitamin D ᶲ			Reference; 69.4% (79/1139)	1.50 (1.13–1.98) ^b^; 10.5% (703/6668) *		
Raisi-Estabragh [38]	Vitamin D						**No difference** in median (±IQR) vitamin D levels, adjusted for seasonality between cases ˠ (33.88 ± 27.01 nmol/L) and non-cases ˠ (35.45 ± 26.78 nmol/L).
Ye [43]	Vitamin D			Reference; 35.6% (36/101) ^#^	3.13 (1.47–6.66); 63.4% (26/41) ^#^		**213% increase in odds of incidence** for those deficient in Vitamin D, compared to those not deficient. **Significant difference in median (IQR) 25(OH)D (nmol/L) levels** between cases ˠ (55.6 (41.9–66.1) nmol/L) and healthy controls (71.8 (57.8–83.7) nmol/L, *p* < 0.05).
Jothimani [26]	Zinc			NR; 33.3% (20/60) ^#^	NR; 84.4% (27/32) ^#^		**Significantly lower median (IQR) zinc levels** cases ˠ (74.5 (53.4–94.6) mg/dL) compared to controls [105.8 (95.65–120.90) mg/dL, *p* < 0.001).

***** between-group *p* < 0.05; ^#^ between group *p*-value not reported; ˠ cases refer to COVID-19 positive individuals, non-cases refer to COVID-19 negative individuals; ᶧ quintiles classification: sufficiency, insufficiency and deficiency; ᶲ quintiles classification: normal and low; ^a^ age-adjusted; ^b^ adjusted for demographic variables, and psychiatric and somatic disorders.

**Table 4 nutrients-13-01589-t004:** COVID-19 Severity defined by without study-specific definitions.

Author [Ref]	Micronutrient in Question	Reported Summary Risk Estimate: Odds Ratio (OR) (95% Confidence Interval)/Mean (SD); % Population Infected (Infected/Population Size)					Key Findings
		Supplementation		Deficiency			
		Supplemented	Non-Supplemented	Quintile 1	Quintile 2	Quintile 3	
Outcome: Death due to COVID-19 episode/during hospitalisation due to COVID-19 episode							
Annweiler [18]	Vitamin D3	0.11 (0.03–0.48) *^,a^; 17.5% (10/57)	Reference; 55.6% (5/9) *				**52% to 97% ^a^ lowered risk of death** in those supplemented with bolus dose, compared to those not supplemented. **Significantly lower** proportion of deceased participants (66.7%) receiving vitamin D3 bolus dose during or just before COVID-19 compared to survivors (92.2%, *p* = 0.023).
Carpagnano [22]	Vitamin D			NR; 3.1% (1/32) ^#^	NR; 20% (2/10) ^#^		**50% probability of dying** in patients with severe vitamin D deficiency after 10 days, compared to those without severe vitamin D deficiency (log-rank test, *p* = 0.019) 5% mortality risk in patients without severe Vitamin D deficiency.
Castillo [47]	Vitamin D	NR; 0% (0/50) ^#^	NR; 7.7% (2/26) ^#^				**Lower proportion** of deaths amongst patients treated with calcifediol, compared to those not treated with calcifediol (*p*-value unreported)
Hastie [17]	Vitamin D			Reference	1.21 (0.83–1.76) ^b^	1.02 (0.75–1.38) ^b^	**2% to 21% ^b^ higher risk of death** in patients with insufficient or deficient Vitamin D levels, compared to those with sufficient Vitamin D levels. **2% decrease in risk of death per 10 nmol/L** increase in Vitamin D levels [Hazard ratio: 0.98 (0.91–1.06) ^b^]
Karahan [27]	Vitamin D3			NR; 0% (0/12) *	NR; 14.7% (5/34) *	NR; 62.1% (64/103) *	**Significantly lower mean ± SD serum 25(OH) vitamin D level** among deceased patients (10.4 ± 6.4 ng/mL) compared to surviving patients (19.3 ± 11.2 ng/mL, *p* < 0.001). **Significantly higher mortality rate** among patients with severe-critical COVID-19 (66.7%) compared with moderate COVID-19 patients (2.1%, *p* < 0.001)
Mardani [31]	Vitamin D						**Significantly lower mean vitamin D concentration** in deceased positive patients (8.175 ng/mL) compared to those surviving positive (19.25 ng/mL) and negative patients (30.17 ng/mL, *p* < 0.0001)
Panagiotou [35]	Vitamin D			Reference; NR	0.97 (0.42, 2.23); NR		**No significant difference** in mortality rates between deficient and normal groups (*p* > 0.05)
Radujkovic [37]	Vitamin D ᶧ			Reference, NR	For all subjects: 14.73 (4.16–52.19) ^c^; NR For inpatients only: 11.51 (3.24–40.92) ^c^; NR		**10.5 ^c^ and 13.7 ^c^ times higher risk of death** in the Vitamin D deficient group, compared to those with not deficient, amongst the inpatients and all patients respectively (*p* < 0.0001)
	Vitamin D ᶲ			Reference, NR	For all subjects: 11.27 (1.48–85.55) ^c^; NR For inpatients only: 7.97 (1.05–60.60) ^c^; NR		**Approximately 7 and 11 ^c^ times higher risk of death** in the Vitamin D insufficient group, compared to those with not insufficient, amongst the inpatients (*p* = 0.004) and all patients (*p* = 0.04)
Tan [49]	Vitamin D, Magnesium & Vitamin B12	NR; 0% (0/17) ^#^	NR; 0% (0/26) ^#^				**No effect** of micronutrient supplementation during hospitalisation on mortality
Capone [46]	Vitamin C & Zinc						Combination of zinc and Vitamin C **supplementation inversely correlated with death incidence** in 73 out of 103 hospitalised patients (71.6%) (Pearson correlation coefficient = −0.10, *p*-value not reported)
Jothimani [26]	Zinc			Reference; 0% (0/20)	5.48 (0.61–49.35); 18.5% (5/27)		**No significant difference in death rates** between zinc-deficient and non-deficient subjects (*p* > 0.05)
Bellmann-Weiler [21]	Iron			Reference; NR	0.458 (0.082–2.572); NR	0.418 (0.15–1.165); NR	**No significant difference in mortality** between patients with absolute or functional iron deficiency, compared to those with no iron deficiency (*p* > 0.05) **13.3 higher odds of mortality** in patients with moderate/severe anemia, compared to whose with no anemia (95% CI: 2.14–83.0, *p* = 0.006)
Wu [41]	Iron (Serum Ferritin)			5.28 (0.72–38.48); NR	Reference; NR		**No significant difference in odds of death** between those with serum ferritin above or below 300 ng/mL, and in median serum ferritin levels between survivors and non-survivors (*p* > 0.05)
Zhao [44]	Iron (Serum iron, Pre-treatment)						**No association between pre-treatment serum iron levels** and odds of mortality in patients (OR: 1.00 (0.98–1.02)_ ^d^) **No significant difference in median pre-treatment serum iron levels** in COVID-19 survivors and non-survivors (median [IQR]: 6.2 µmol/L [4.3–8.0] vs. 4.1µmol/L [2.2–7.5], *p* > 0.05)
	Iron (Serum iron, Post-treatment)						**Up to 2% lowered odds of mortality** with higher post-treatment serum iron levels (OR: 0.99 (0.98–1.00_ ^d^) **Significantly higher median post-treatment serum iron levels in COVID-19 survivors,** compared to non-survivors (median [IQR]: 19.1 µmol/L [13.2–25.6] vs. 5.5 µmol/L [3.5–11.1], *p* = 0.002)
Zhou [45]	Iron (Serum Ferritin)			9.1 (2.04–40.58); 43.1% (44/102) *	Reference; 7.7% (2/26) *		**8.1 times higher odds of death** amongst those with serum ferritin >300 ng/mL, compared to those ≤300 ng/mL (*p* = 0.0038) **Significantly higher median (IQR) serum ferritin levels** (μg/L) in survivors (503.2 (264.0–921.5) μg/L) and non-survivors (1435.3 (728.9–2000.0) μg/L, *p* < 0.0001)
Moghaddam [34]	Selenium (Serum)			NR; 13.0% (12/92) ^#^	NR; 29.7% (22/74) ^#^		**Significantly higher mean ± SD serum levels** in survivors (53.3 ± 16.2 μg/L) than in non-survivors (40.8 ± 8.1 μg/L, *p* < 0.001)
Outcome: Hospitalisation Duration (Unit: days)							
Carpagnano [22]	Vitamin D			12.5 (8–20.5) ^#,f^	8 (6–11.25) ^#,f^		**Shorter median length of stay** in respiratory intermediate care unit for patients with severe deficiency (8 days), compared to those with non-severe deficiency (12.5 days, *p*-value not reported), as the former tend to experience death or transfer to intensive care units.
Maghbooli [30]	Vitamin D3			5 (1–19) ^g^	5 (1–23) ^g^		**No difference in median length of stay** between those with and without deficiency or insufficiency (*p* > 0.05).
Jothimani [26]	Zinc			5.7 (NR) *	7.9 (NR) *		**Significantly longer mean length of stay in patients with deficiency** (7.9 days), compared to those with no deficiency (5.7 days, *p* = 0.048) **239% higher odds of hospitalisation****≥****7 days** for deficient patients, compared to non-deficient patients (OR: 3.39 (95% CI: 0.99–11.57)) **Significantly higher proportion of patients with deficiency hospitalised****≥****7 days** (59.3%, 16/27), compared to those without deficiency (30%, 6/20; *p* = 0.047)
Outcome: Intensive care unit (ICU) Admission							
Bellmann-Weiler [21]	Iron			Reference; NR	0.147 (0.017–1.297); NR	0.556 (0.225–1.373); NR	**No significant difference in ICU admission rates** between patients with absolute or functional iron deficiency, and those without iron deficiency
Carpagnano [22]	Vitamin D			NR; 12.5% (4/32) ^#^	NR; 20% (2/10) ^#^		**Similar ICU admission rates** between patients with and without severe deficiency (*p*-value not reported)
Castillo [47]	Vitamin D	0.03 (0.003–0.25) ^e^; 2% (1/50) *	Reference; 50% (13/26) *				**97% reduction in odds for ICU admission** amongst inpatients treated with calcifediol, compared to those not given calcifediol **Significantly lower proportion of ICU admissions amongst inpatients treated with calcifediol**, compared to those not given calcifediol (*p* < 0.001)
Maghbooli [30]	Vitamin D3			NR; 14.3% (11/77)	NR; 20.9% (33/158)		**No significant difference in ICU admission rates** between those with and without deficiency or insufficiency (*p* > 0.05).
Panagiotou [35]	Vitamin D			NR; 18.2% (8/44) *	NR; 37.8% (34/90) *		**No significant difference in mean difference for logarithmically transformed-25(OH)D levels** between patients admitted and not admitted to the ITU (mean difference: 0.14; 95% CI: −0.15–0.41, *p* = 0.3), although those admitted to the ITU had lower 25(OH)D levels (33.5 nmol/L ± 16.8) than those not admitted (48.1 nmol/L ± 38.2) **Significantly higher vitamin D deficiency prevalence in patients admitted to ITU** (81%), compared to those not admitted to the ITU (60.9%, *p* = 0.02).
Jothimani [26]	Zinc			Reference; 10% (2/20)	3.15 (0.58–17.67); 25.9% (7/27)		**No significant difference in odds of ICU admission** between patients with and without deficiency **No significant difference in ICU admission rates** between patients with and without deficiency (*p* > 0.05)

***** between-group *p* < 0.05; ^#^ between group *p*-value not reported; ᶧ quintiles classification: no deficiency and deficiency; ᶲ quintiles classification: no insufficiency and insufficiency; ^a^ Hazard ratio adjusted for recent bolus vitamin D3 supplementation; ^b^ Hazard ratio adjusted for age, sex, ethnicity, month of assessment, Townsend deprivation quintile, household income, BMI category, smoking status, diabetes, systolic blood pressure, diastolic blood pressure, self-reported health rating, and long-standing illness, disability or infirmity; ^c^ Hazard ratio adjusted for age, gender, and comorbidities for all following; ^d^ Adjusted for age, lymphocyte percentage, lymphocyte count, pretreatment serum iron level, and posttreatment serum iron level; ^e^ Adjusted for hypertension and type 2 diabetes; ^f^ Median (IQR); ^g^ Median (range).

**Table 5 nutrients-13-01589-t005:** COVID-19 Severity indicated by outcomes with study-specific definitions.

Author [Ref]	Micronutrient in Question	Outcome Definition (Study-Specific)	Reported Summary Risk Estimate: Odds Ratio (OR) (95% Confidence Interval); % Population Infected (Infected/Population Size)					Key Findings
			Supplementation		Deficiency			
			Supplemented	Non-supplemented	Quintile 1	Quintile 2	Quintile 3	
Outcome: Clinical severity								
Hastie [17]	Vitamin D	Hospitalisation rate			Reference; NR	1.06 (0.89–1.26) ^a^; NR	1.1 (0.88–1.37) ^a^; NR	**No significant difference in hospitalisation rates** between those with insufficient or deficient Vitamin D, compared to those with sufficient Vitamin D. **No significant difference in hospitalisation rates for every 10 nmol/L increase** in Vitamin D levels (Incidence rate ratio: 1.00 (0.96–1.05) ^a^)
Merzon [33]	Vitamin D	Hospitalisation rate						**Significantly lower mean plasma 25(OH)D levels in hospitalised patients** (18.38 ng/mL (95% CI: 16.79–19.96)), compared to non-hospitalised patients. 20.45 ng/mL (95% CI: 20.22–20.68), *p* < 0.001).
Radujkovic [37]	Vitamin D	Hospitalisation rate			NR; 44.4% (64/144) *	NR; 70.7% (29/41) *		**Significantly higher proportion of hospitalised patients in those with deficiency** than those with no deficiency (*p* = 0.004) **Significantly different median (IQR) Vitamin D levels** between inpatients (14.6 ng/mL (10.7–21.0)) and outpatients (18.6 ng/mL (14.2–26.0), *p* = 0.001)
Karahan [27]	Vitamin D3	Chinese Clinical Guideline (Moderate disease)			NR; 100% (12/12) *	NR; 79.4% (27/34) *	NR; 7.8% (8/103) *	**Significantly different proportions** of patients with moderate and severe/critical disease onset across those with normal, insufficient and deficient Vitamin D3 levels (*p* < 0.001)
		Chinese Clinical Guideline (Severe/critical disease)			NR; 0% (0/12) *	NR; 18.9% (7/37)*	NR; 92.2% (95/103)	
Maghbooli [30]	Vitamin D3	CDC criteria (Severe/critical disease)			Reference; 63.6% (49/77) *	1.59 (1.05–2.41) *^,b^; 77.2% (122/158) *		**59% ^b^ higher odds of severe/critical disease** amongst patients with deficiency or insufficiency, compared to those without deficiency **Significantly higher proportion of severe/critical onset** amongst deficient or insufficient patients compared non-deficient patients (*p* = 0.02)
Ye [43]	Vitamin D	Chinese National Health Commission Guidelines (6th edition) (Mild/moderate disease)			Reference; 88.9% (32/36) ^#^	NR; 69.2% (18/26) ^#^		**14.18 ^c^ times higher odds for severe/critical COVID-19** amongst patients with vitamin D deficiency compared to those without deficiency (*p* = 0.034) **Significantly higher median (IQR) 25(OH)D (nmol/L) levels** in mild/moderate patients (56.6 nmol/L (44.6–66.4); *n* = 50 patients), compared to severe/critical patients (38.2 nmol/L (33.2–50.5); *n* = 10 patients, *p* < 0.05)
		Chinese National Health Commission Guidelines (6th edition) (Severe/critical disease)			Reference; 5.6% (2/36) ^#^	15.18 (1.23–187.45) ^c^; 30.8% (8/26) ^#^		
		Number of symptomatic patients			NR; 80% (8/10)	NR; 100% (26/26)	NR; 100% (26/26)	**Significantly different proportions** of symptomatic patients across those with sufficient, insufficient and deficient Vitamin D levels (*p* = 0.004)
Pizzini [36]	Vitamin D	Requiring hospitalisation, respiratory support or intensive care treatment						**No significant difference in mean ± SD 25(OH)D levels**, 8 weeks after disease onset, across patients with varying COVID-19 severity: overall 54 ± 25 nmol/L, mild 64 ± 31 nmol/L, moderate 54 ± 19 nmol/, and severe 50 ± 24 nmol/L (*p* > 0.05)
	Calcium (total, ionised)							**No significant difference in mean ± SD total calcium levels**, 8 weeks after disease onset, across patients with varying COVID-19 severity: overall 2.37 ± 0.09 mmol/L, mild 2.37 ± 0.09 mmol/L, moderate: 2.36 ± 0.09 mmol/L, and severe: 2.39 ± 0.08 mmol/L (*p* > 0.05) **No significant difference in mean ± SD ionised calcium levels**, 8 weeks after disease onset, across patients with varying COVID-19 severity: overall 1.22 ± 0.04 mmol/L, mild: 1.24 ± 0.03 mmol/L, moderate: 1.22 ± 0.04 mmol/L, and severe: 1.22 ± 0.04 mmol/L (*p* > 0.05)
	Iron (Serum Ferritin)							**Significantly different mean ± SD serum ferritin levels**, 8 weeks after disease onset, across patients with varying COVID-19 severity: overall 263 ± 230 µg/L, mild: 139 ± 118 µg/L, moderate: 260 ± 183 µg/L, and severe: 317 ± 271 µg/L (*p* = 0.001)
Dahan [24]	Iron (Serum Ferritin)	Report of the WHO-China Joint Mission						**Significant difference in mean serum ferritin levels** across disease severity groups: mild 327.27 ng/mL, moderate 1555 ng/mL, severe 2817.6 ng/mL (*p* = 0.003). Significantly higher mean serum ferritin levels, in the moderate and severe disease groups, compared to the mildly ill group (*p* = 0.006 and 0.005, respectively). **No significant difference between the moderate and severe disease groups** ferritin levels (*p* > 0.05) after excluding extremely deviant cases. **Significantly higher mean serum ferritin levels in severe patients** (2817.6 ng/mL), than in non-severe patients (708.6 ng/mL, *p* = 0.02).
Sun [19]	Iron (Serum Ferritin)	New Coronavirus Pneumonia Prevention and Control Program, 7th edition						**Significant difference in serum ferritin levels (times the upper limit of the normal)** between the mild and critically ill groups, and between the moderate disease and critically ill groups (*p* < 0.01); Serum ferritin levels, mean ± SD (times the upper limit of the normal)—mild: 0.55 ± 0.5, moderate: 2.00 ± 2.20, severe: 3.20 ± 1.47, critically ill: 5.08 ± 3.29
Zhao [44]	Iron (Serum)	Chinese National Health Commission Guidelines (7th edition)						**Significantly lower median pre-treatment serum iron levels in patients with severe COVID-19** compared to those with mild COVID-19 (*p* < 0.05). **No significant difference in median(IQR) pre-treatment serum iron levels across all groups**—mild: 6.6 μmol/L (5.4–10.9), severe: 4.9 μmol/L (4.0–8.1), critical: 5.2 μmol/L (2.6–7.3) (*p* > 0.05).
Smith [39]	Iron (Ferritin)	Hospitalisation and/or ICU admission, requiring mechanical ventilation and/or death						**Significantly higher median (IQR) ferritin levels in severe patients** (1163 ng/mL (640.0–1967.0)) compared to those with moderate condition (624.0 ng/mL (269.7–954.0), *p* < 0.01)
Sonnweber [40]	Iron/Ferritin	ICU admission, requiring oxygen therapy or respiratory support						**No significant difference in mean ± SD iron levels between severity groups**: mild 18 ± 6 µmol/L, moderate:16 ± 6 µmol/L, and severe/critical 15 ± 6 µmol/L (*p* > 0.05). **Significantly different mean ± SD ferritin levels between severity groups**: mild 139 ± 118 µmol/L, moderate 260 ± 183 µmol/L, and severe/critical 317 ± 271 µmol/L (*p* = 0.001).
Yasui [42]	Iron (Ferritin)	ICU admission, requiring oxygen therapy or respiratory support			All patients: NR; 36.7% (11/30) * Subset of inpatient: NR; 28.6% (6/21)	All patients: NR; 6.3% (2/32) * Subset of inpatient: NR; 12.5% (1/8)		**Significantly higher proportion of severe patients** among those with ≥300 ng/mL (males)/≥200 ng/mL (females) ferritin levels, compared to those with ferritin levels lower than that, amongst all patients (*p* = 0.008) **Significantly higher mean ± SD ferritin levels in severe patients** (1117 ± 654 ng/mL), compared to the mild/moderate patients (386 ± 393 ng/mL, *p* = 0.00002)
	Zinc				Subset of inpatient: NR; 5% (1/20) *	Subset of inpatient: NR; 66.7% (6/9) *		**Significantly higher proportion of severe patients among those with****≥****70****µ****g/dL zinc levels**, compared to those with zinc levels lower than that, amongst all inpatients (*p* = 0.0003) **Significantly lower mean ± SD zinc levels** in severe patients (62.4 ± 19.2 μg/dL), compared to mild/moderate patients (87.7 ± 19.1 μg/dL, *p* = 0.005)
Jothimani [26]	Zinc	Number of symptomatic patients			Reference; 90% (18/20)	3.15 (0.58–17.67); 96.3% (26/27)		**No association between zinc deficiency and odds of symptoms onset** **No significant difference in proportion of symptomatic patients** between patients with and without zinc deficiency (*p* > 0.05)
Outcome: Progression to respiratory-related complication								
Jothimani [26]	Zinc	ARDS development			NR; 0% (0/20)	NR; 18.5% (5/27)		No significant difference proportion of patients developing ARDS between those with and without zinc deficiency (*p* > 0.05)
Wu [41]	Iron (Serum Ferritin)	ARDS development			3.53 (1.52-8-16); NR	Reference; NR		253% higher odds of ARDS development in those with serum ferritin levels >300 ng/mL, compared to those with levels ≤300 ng/mL (*p* = 0.003) Significantly higher median serum ferritin levels in patients with ARDS, compared to those without ARDS (difference: 545.5 ng/mL (IQR 332.15–754.44), *p* < 0.001)
Maghbooli [30]	Vitamin D3	ARDS development			NR; 11.7% (9/77)	NR; 17.1% (27/158)		No significant difference in patients developing ARDS between those with and without Vitamin D deficiency (*p* > 0.05)
Im [25]	Vitamin D3	Pneumonia incidence, or requiring high-flow nasal cannula, mechanical ventilator, and extracorporeal membrane oxygenation or death			NR; 50% (6/12) ^#^	NR; 68.4% (26/38) ^#^		Similar proportions of patients with outcomes between those with deficiency and no deficiency (*p* = not reported) About half as many patients with Vitamin B6 deficiency developing respiratory complications compared to patients without deficiency (*p* = not reported)
	Vitamin B6				NR; 66% (31/47) ^#^	NR; 33.3% (1/3) ^#^		
	Vitamin B9				NR; 64.6% (31/48) ^#^	NR; 50% (1/2) ^#^		
	Selenium				NR; 65.5% (19/29) ^#^	NR 61.9% (13/21) ^#^		
	≥1 deficiency				NR; 44.4% (4/9) ^#^	NR; 68.3% (28/41) ^#^		
Capone [46]	Vitamin C & Zinc	Requiring invasive mechanical ventilation						Requiring invasive mechanical ventilation inversely correlated with zinc and vitamin C supplementation amongst 73 hospitalised patients (Pearson correlation coefficient = −0.20)
Radujkovic [37]	Vitamin D	Requiring any form of oxygen therapy			NR; 35.4% (54/144) *	NR;63.4% (26/41) *		Significantly higher proportion of deficient patients requiring any form of oxygen therapy compared to non-deficient patients (*p* < 0.001)
Tan [49]	Vitamin D, Magnesium & Vitamin B12	Requiring oxygen therapy	0.195 (0.041–0.926) ^d^, 0.182 (0.038–0.859) ^e^; 11.7% (2/17) *	Reference; 30.8% (8/26) *				80.5% ^d^ to 81.8% ^e^ lowered odds of requiring oxygen therapy amongst patients provided with micronutrient blend, compared to those not provided with it. Significantly higher proportion of patients not provided with micronutrient blend requiring oxygen therapy, compared to those provided with micronutrient blend (*p* = 0.006)
Outcome: Composite outcome (with multiple outcomes)								
Liu [28]	Calcium (Serum calcium)	Need for mechanical ventilation, ICU admission due to COVID-19 episode, or all-cause mortality during admission			Reference; 25% (10/40) *	2.962 (1.085–8.090) ^f^; 47.8% (32/67) *		196% ^f^ higher odds for composite outcome in those with hypocalcemia compared to those with normal serum calcium levels. Significantly lower median (IQR) serum calcium levels in patients with composite outcome (2.01 mmol/L (1.97–2.05)) compared to those without the outcome (2.10 mmol/L (2.03–2.20), *p* < 0.001).
Macaya [29]	Vitamin D3 ^α^	Death, ICU admission or requiring high flow oxygen (greater than nasal cannula)			Reference; 31.4% (11/35)	3.2 (0.9–11.4) ^g^; 44.4% (20/45)		220% ^g^ higher odds for death, ICU admission or requiring oxygen in those with deficiency, compared to those without deficiency Patient age significantly modified the association (*p*_interaction_ = 0.03) Age over 75 years (3rd tertile) and male gender were significantly associated with the composite outcome [OR 10.4 (95% CI: 2.0–54.8) vs. the first tertile, *p* = 0.006; OR 6.2 (95% CI: 2.0–19.5), *p* = 0.002, respectively]. No significant difference in the median (IQR) levels of 25(OH)D between patients with outcome (13 ng/mL (8–25)) and without the outcome (19 ng/mL (9–30), *p* > 0.05)
	Vitamin D3 ^β^		NR; 45.5% (20/44)	NR; 20.6% (11/36)				No significant difference in proportion of patients with composite outcome, between those supplemented or not supplemented (*p* > 0.05)
Radujkovic [37]	Vitamin D3 ᶧ	Mechanical invasive ventilation and/or death from COVID-19 episode			For all subjects: Reference; NRFor inpatients only: Reference; 87.5% (56/64) ^#^	For all subjects: 6.12(2.79–13.42) ^h^; NR For inpatients only: 4.65 (2.11–10.25) ^h^; 89.7% (26/29) ^#^		365% ^h^ and 512% ^h^ higher hazards for composite outcome in those with vitamin D deficiency, compared to those without deficiency in the inpatient and all patients respectively. Significantly different proportion of patients receiving different types of maximum oxygen therapy (including none) differed significantly between the deficiency and non-deficiency group, for entire cohort (*p* < 0.001) and inpatient subgroup (*p* = 0.004)
	Vitamin D3 ᶲ				Reference; NR	For all subjects: 5.75 (1.73–19.09) ^h^; NR For inpatients only: 3.99 (1.2–13.28) ^h^; NR		299% ^h^ and 475% ^h^ higher hazards for composite outcome in those with vitamin D insufficiency, compared to those without insufficiency in the inpatient and all patients respectively.
Tan [49]	Vitamin D, Magnesium & Vitamin B12	Requiring oxygen therapy or ICU admission due to COVID-19 episode	NR; 5.9% (1/17) *	NR; 30.8% (8/26) *				Significantly higher proportion of composite outcome in those not provided with the micronutrient blend, compared to those provided with the blend (*p* = 0.006)

***** between-group *p* < 0.05; ^#^ between group *p*-value not reported; ^α^ Exposure was deficiency assessed as an ordinal variable (Yes/No); ^β^ Exposure was supplementation status assessed as an ordinal variable (Yes/No); ᶧ quintiles classification: no deficiency and deficiency; ᶲ quintiles classification: no insufficiency and insufficiency; ^a^ Incidence rate ratio adjusted for age, sex, ethnicity, month of assessment, Townsend deprivation quintile, household income, BMI category, smoking status, diabetes, systolic blood pressure, diastolic blood pressure, self-reported health rating, and long-standing illness, disability or infirmity; ^b^ Adjusted for age, sex, BMI, smoking and history of a chronic medical disorder; ^c^ Adjusted for age (every 10 years), gender, renal failure, diabetes, and hypertension; ^d^ Adjusted for age; ^e^ Adjusted for hypertension; ^f^ Adjusted for age, and c-reactive protein, procalcitonin, interleukin-6 and D-dimer levels; ^g^ Adjusted for age, gender, obesity and severe chronic kidney disease; ^h^ Hazard ratio adjusted for age (≥60 years), gender and comorbidity status (Yes/No).

## Data Availability

The data presented in this study are available on request from the corresponding author.

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
