# Peer review of "Micronutrients Deficiency, Supplementation and Novel Coronavirus Infections—A Systematic Review and Meta-Analysis"

_nutrients, 2021, doi:10.3390/nu13051589_

Round 1

Reviewer 1 Report

The manuscript is an interesting systematic review and meta-analysis about the micronutrient deficiency in covid-19 patients. This is a timely topic and important research and clinical aspect. The paper is well written and the methodology is well explained. I have only a concern about the methodology that the authors should take into consideration. The database search was performed in October 2020, 6 months ago. This is a really huge period for a systematic review and meta-analysis on a topic that is so recent (and authors have found a small number of papers). Why the authors did not perform an update of the search in the meantime? 

Author Response

Dear Reviewer, we thank you for your kind comments! Although the current search yielded a relatively smaller number of studies, considerable time and effort was dedicated to assess the methodological quality of all 33 studies included in the review using the appropriate tools, and subsequently pool and explore the results through a variety of clinically meaningful proxy outcomes for severity. Thus, the 6 months taken to complete the review is reasonable, as literature regarding any disease typically do not typically change that quickly due to varying levels of research interest. In addition, the current search covered a period of 8 months between Feb and Oct 2020, and should provide adequate evidence of micronutrients’ effect on COVID-19.

Nonetheless, we acknowledge that COVID-19 is a rapidly evolving pandemic of high global interest, resulting in the daily generation of a large volume of related literature. We also concur with your view that that an update to the literature is important to provide a comprehensive overview of available evidence. Thus, we updated the search on 26 April 2021 and found a total of 1,326 new titles using the same search strategy we used in October 2020. In light of the large number of new literature generated in this period, we believe that it may be more appropriate that the search update is included as a separate follow-up project. This is due to the time and effort required to screen and further assess the new literature, if they are deemed eligible for inclusion. In addition, the new literature is also highly likely to be focused on COVID-19, and thus it might be more appropriate that the update is focused on COVID-19 for specificity, instead of general novel coronavirus infections as is the focus of this review.

Reviewer 2 Report

Dear authors,

The study is quite interesting and easy to read and to understand. Congratulation for the hard work reviewing and included all that information about the founded studies. I have some questions and suggestions.

  • The abstract have some contradictory information. First you say that "there was a significant reduction on odds of COVID-19 ICU admissions or severe/critical disease onset (pooled 23 OR: 0.26, 95% CI: 0.08, 0.89)" But after that you indicate that "Insignificant protective effects were observed on other outcome  mortality, ICU admission severe/crit". I may do not understand the meaning but I think is contrary.
  • The introduction is well written but I miss some information about the micronutrients defficiency and supplementation in other kind of viruses.
  • The study aim is clear and adequate for a review and meta-analysis.
  • You included some meta-analysis with one study. For example with "Zinc", "Selenium", "Zinc" again, "Ferritin" etc. Please delete that from the figures and the results.
  • Tables are quite clear and includes good information, congratulations.
  • The discussion and conclusion are well written and easy to understand and include good information.
  • Kind regards

Author Response

The study is quite interesting and easy to read and to understand. Congratulation for the hard work reviewing and included all that information about the founded studies. I have some questions and suggestions.

  1. The abstract have some contradictory information. First you say that "there was a significant reduction on odds of COVID-19 ICU admissions or severe/critical disease onset (pooled 23 OR: 0.26, 95% CI: 0.08, 0.89)" But after that you indicate that "Insignificant protective effects were observed on other outcome mortality, ICU admission severe/crit". I may do not understand the meaning but I think is contrary.

Thank you for your comment. We would like to clarify that the first sentence (“there was a significant reduction on odds of COVID-19 ICU admissions or severe/critical disease onset (pooled 23 OR: 0.26, 95% CI: 0.08, 0.89)") was referring to the protective effect of micronutrients on the composite outcome for severity, when the event (i.e. severe COVID-19 episode) was defined as either ICU admissions or severe/critical disease onset. In the subsequent sentence (“Insignificant protective effects were observed on other outcome mortality, ICU admission severe/crit”), we were referring to the protective effect of micronutrients on the following outcomes for severity, when the event was defined as either 1) mortality, 2) ICU admission, 3) progression to respiratory-related complications, 4) severe/critical disease onset or requiring respiratory support and hospitalization rate.

We hope that this helped clarify your doubt, and have also slightly revised the phrasing in the abstract in lines 23-24 (“and ICU admissions or severe/critical disease onset combined as a severity outcome (pooled OR: 0.26, 95% CI: 0.08, 0.89).”) to better reflect that the first sentence was referring to a combined outcome.

  1. The introduction is well written but I miss some information about the micronutrients defficiency and supplementation in other kind of viruses.

Thank you for your kind insight. We have included the information about the effect of micronutrient deficiency and supplementation on other respiratory diseases with a similar mode of transmission, such as upper respiratory tract infections, in lines 528 to 535 the discussion section. Thus we did not originally include it in the introduction to keep the writing concise and prevent repetition.

However, at your suggestion we have included some information about its effect on acute respiratory tract infections, colds and influenza in lines 62 to 65 in the introduction in the revised manuscript.

  1. You included some meta-analysis with one study. For example with "Zinc", "Selenium", "Zinc" again, "Ferritin" etc. Please delete that from the figures and the results.

Thank you for your kind comment, and feel free to correct us if we interpreted what you meant wrongly. We would like to clarify that we included these single studies in the forest plots for the purpose of exploring the effect of micronutrients in general, regardless of its identity, on the outcome of interests. For each outcome, the studies were stratified according to micronutrients to 1) be transparent and 2) provide readers with the composition underlying each forest plot, and 3) inform readers on the potential effect of each micronutrient should there be sufficient studies. While the effect estimates of each micronutrient were shown in the forest plots, we did not discuss the effect of the micronutrient singly in prose, should there only be a single study for that micronutrient due to concerns about statistical accuracy. This review only reported the combined effect of all micronutrients for each outcome in its entirety. We apologise for the confusion caused, and hope this above explanation clarifies your confusion.  

  1. The study aim is clear and adequate for a review and meta-analysis.
  2. Tables are quite clear and includes good information, congratulations.
  3. The discussion and conclusion are well written and easy to understand and include good information.

Thank you for your kind comments!

Round 2

Reviewer 2 Report

Dear authors,

Thank you for addressing the required changes.

I understand what you mean about including just one study in the forestplots in some variables but it will be confusing for most of the readers so please delete those analysis from the forestplots and if you want to show that information about one study you can create one table with all the information for variables with only one study.

Kind regards

Author Response

Dear Reviewer, thank you for your concern and kind suggestion to prevent confusing the reader. We have since removed the stratification by micronutrients in all forest plots, and instead included the micronutrient identity in brackets beside each study included to address your concern and concurrently provide transparency to the reader, with respect to the micronutrient being assessed in the respective studies and forest plot. We hope this addresses your concern about confusing the readers. Thank you.